# Trends in the prevalence of obesity and estimation of the direct health costs attributable to child and adolescent obesity in Brazil from 2013 to 2022

Eduardo Augusto Fernandes Nilson[1,2,3], Michele Gonçalves da Costa[4], Ana Carolina Rocha de Oliveira[4]*, Olivia Souza Honorio[4], Raphael Barreto da Conceição Barbosa[4]

1 Oswaldo Cruz Foundation- Fiocruz, Brasilia, Brazil, 2 Center for Epidemiological Research in Nutrition and Public Health, University of São Paulo, São Paulo, Brazil, 3 Universidad Autónoma de Chile, Santiago, Chile, 4 Desiderata Institute, Rio de Janeiro, Brazil

* carolina.rocha@desiderata.org.br

## Abstract

### Introduction

Childhood obesity is a major global public health issue globally and in Brazil. The impacts of childhood obesity include higher risk of disease during childhood and of obesity and non-communicable diseases in adulthood and represent an important epidemiological and economic burden to countries. This study aims to analyze the trends and to estimate the direct healthcare costs of childhood and adolescent obesity to the National Health System from 2013 to 2022.

### Methods

We used Prais-Winsten regressions for determining the trends in the prevalence of obesity and modeled the attributable to childhood and adolescent obesity in the Brazilian National Health System using previous meta-analysis of studies.

### Results

The hospitalizations of children and adolescents with obesity as a primary cause totaled Int $2.6 million to the Brazilian National Health System from 2013 to 2022, demonstrating that obesity is rarely considered as a cause of hospitalization especially among children and adolescents. The additional costs of hospitalizations attributable to childhood obesity totaled Int$101.5 million during the same period. The additional non-hospital, outpatient and medication cost attributable to childhood obesity in Brazil were estimated at Int$6.0 million, so the total estimated healthcare costs were of approximately Int$107.5 million in the last decade.

**Data Availability Statement:** All relevant data are within the manuscript and its Supporting Information files.

**Funding:** The author(s) received no specific funding for this work.

**Competing interests:** We have read the journal's policy and the authors of this manuscript have the following competing interests: EAFN is currently an Academic Editor for PLOS One. Nevertheless, this does not alter our adherence to PLOS ONE policies on sharing data and materials.

## Conclusion

This study highlights that childhood and adolescent obesity are increasing for most age-groups and that its costs are not limited to the economic impacts on adult health and represent a relevant economic burden to the Brazilian National Health System and to families because of additional costs during childhood and adolescence. Therefore, the prevention and control of childhood and adolescent obesity must be public health priorities.

## Introduction

Childhood and adolescent obesity currently represent major public health challenges. Globally, in 2020, there were approximately 175 million children over 5 years of age and adolescents with obesity and, if current trends are not modified, this number may increase to 383 million children and adolescents by 2035 [1].

Despite the lack of regular national nutritional surveys on child nutrition over the last decade, the prevalence of obesity among Brazilian children under 12 years of age was estimated at 12.2% in a recent systematic review [2]. Additionally, the prevalence of obesity among children from 5 to 9 years of age accompanied by public Primary Health Care facilities reached 13,2% in 2018 [3]. For adolescents, national surveys showed a gradual increase in the prevalence of overweight and obesity over the last decades [4].

A growing body of evidence has demonstrated that, compared to children with adequate weight, children with obesity have a higher risk of health issues during childhood, such as type-2 diabetes, hypertension, asthma, sleep apnea, musculoskeletal problems and metabolic disorders, as well as lower self-esteem, higher risk of bullying and lower school performance [5]. Additionally, children and adolescents with obesity face higher risk of health issues in adulthood and childhood obesity is a string predictor of adult obesity and risk of non-communicable diseases such as type-2 diabetes, cardiovascular diseases, and some types of cancers. The future impacts of childhood obesity also include negative socioeconomic and workforce consequences, including reduced employability, lower productivity and salaries [6].

Considering this association between obesity during childhood and adolescence and negative outcomes in adulthood, many studies have started to project the impacts of childhood obesity on morbidity and mortality associated with comorbidities and the direct costs of treating these diseases and also the indirect costs to society related to premature mortality and retirement and losses of productivity due to absenteeism and presenteeism [7–11].

Despite inclusive evidence on the increased risk of hospitalization among children and adolescents with obesity, strong evidence has demonstrated that childhood obesity is associated with higher risk of visits to emergency rooms [12,13] and higher risks of hospitalization and acute infection by diseases such as influenza e Covid-19 [14,15].

A number of studies showed that children and adolescents with obesity have higher costs of hospitalizations, outpatient procedures, prescription medications and non-hospital costs [5,16–20]. Based on these primary studies, a recent systematic review and meta-analysis compared the healthcare costs of children and adolescents with obesity to those with normal BMI (body-mass index) and found increased annual total medical costs attributable to childhood overweight and obesity of $237.55 per capita and annual direct and indirect costs of $13.62 billion, which were projected to reach $49.02 billion in 2050 [21].

In Brazil, previous studies have estimated the economic burden of obesity-related diseases among adults to the National Health System (Sistema Único de Saúde–SUS) [22,23], however,

there is a need to develop studies on the attributable to childhood and adolescent obesity. Therefore, this study aims to fill a knowledge gap regarding the economic burden of child and adolescent obesity in Brazil. The objectives of this study are to analyze the trends in the prevalence and to estimate the direct healthcare costs of childhood and adolescent obesity to the National Health System from 2013 to 2022.

## Methods

This study is based on the estimation of the trends in the prevalence of childhood and adolescent obesity and the modeling of total costs of hospitalizations and the additional costs attributable to childhood and adolescent obesity in the Brazilian National Health System from 2013 to 2022. For the purposes of this study, because of the age disaggregation available in the public reports of the national health information systems, we considered individuals from 0 to 9 years of age as children and those aged 10 to 19 years as adolescents [24].

The modeling is divided in the following steps: (I) fitting the prevalence of childhood and adolescent obesity from 2013 to 2022 through a linear regression analysis; (II) estimating the percentage of additional hospitalization costs of children and adolescents with obesity compared to those without obesity; (III) extracting the data on the total number and average per capita costs of hospitalizations; (IV) extracting and adjustment of data on the prevalence of obesity among children and adolescents that were monitored by Primary Healthcare services; (V) estimating the number of hospitalizations and the average per capita cost of hospitalizations among children and adolescents with obesity and those with adequate BMI; (VI) estimating the total costs of hospitalizations of children and adolescents with obesity and the additional costs compared to those with adequate BMI.

### Trends in child obesity

The information on the prevalences of obesity among individuals monitored by the public primary healthcare services by age group (0 to 4 years, 5 to 9 years, and 10 to 19 years) for each year from 2012 to 2022 were obtained from publicly available reports from the Food and Nutrition Information System–Sisvan [25].

The trends in the changes in the prevalence of obesity among children and adolescents were estimated through the Prais-Winsten method [26] for each age group.

### Cost analysis

Firstly, before estimating the total costs of hospitalizations of children and adolescents with obesity and the additional costs attributable to childhood obesity, we extracted the hospitalization costs which primary cause was declared as obesity in the National Hospital Information System from 2013 to 2022 by age and sex groups to analyze how childhood obesity is considered a disease by health professionals in the National Health System.

Then, as the first step for estimating the hospitalization costs attributable to childhood and adolescent obesity, we calculated the average percentage of additional hospitalization costs of children and adolescents with obesity compared to those with adequate BMI based on a selection of the 24 primary studies on the healthcare costs of childhood and adolescent obesity that were included in the meta-analysis by Ling et al, 2023 [21]. The studies had variable sample sizes and included children and adolescents from 0 to 19 years of age, and the estimated percentage of additional hospitalization costs of children and adolescents with obesity was of 16.46% (CI 95%: 1.98%-30.94%), as detailed in the Supplementary Materials.

**Table 1. Data inputs for the estimation of time trends in childhood and adolescent obesity and of its direct healthcare costs.**

| Parameter | Outcome | Comments | Source |
|---|---|---|---|
| Childhood obesity | Prevalence of children and adolescents with obesity (%) | Stratified by age and sex | National Food and Nutrition Surveillance System–Sisvan [25] |
| Additional healthcare costs attributable to childhood obesity | Percentage of additional healthcare costs among children and adolescents with obesity (%) | | Ling et al, 2023 [21] |
| Number of hospitalizations | Total hospitalizations of children and adolescents from all-causes in the National Health System (number) | Stratified by year and age-group | National Hospital' Information System (SIH/SUS) [27] |
| Costs of hospitalizations | Costs of hospitalizations from all-causes to the National Health System (Int$) | Stratified by year, age and sex | National Hospital' Information System (SIH/SUS) [27] |

This approach was used because there are no available studies comparing the costs of hospitalization between children and adolescents with obesity and those with adequate BMI in Brazil. This methodological approach also intends to avoid directly multiplying the estimated costs found in the meta-analysis because of the differences considering the organization and coverage of health systems and because all studies included in the meta-analysis were conducted in high income countries, therefore the attributable costs could be overestimated if the absolute values were considered.

Afterwards, we extracted the average per capita costs for all hospitalizations and the total number of hospitalizations from 2013 to 2022 for each age group (1 to 4 years, 5 to 9 years, 10 to 14 years, and 15 to 19 years) from the National Hospital Information System–SIH/SUS [27] and converted the currency using the Purchasing Power Parities estimated by the Organisation for Economic Co-operation and Development [28].

The trends on the nutritional status of children and adolescents (obese and non-obese) were considered as proxies of the nutritional status of the population that predominantly uses the public health system.

We conservatively assumed that children and adolescents with obesity had the same risk of hospitalization from all causes of those with adequate BMI to estimate the number of hospitalizations among children and adolescents with obesity based on the prevalence of obesity by age-group for each year of the analysis.

The additional costs attributable to childhood and adolescent obesity were estimated by multiplying the additional average costs and the total number of hospitalizations of children and adolescents with obesity.

Finally, the additional non-hospital costs, together with outpatient and medication costs were estimated assuming that they would keep the proportionality to the additional hospitalization costs estimated by Ling et al [21].

All data used in the modeling study are summarized in Table 1.

## Sensitivity analysis

The robustness of the model assumptions and its data inputs was assessed through sensitivity analysis changing the original model parameters and comparing the results to the primary model. The modeled scenarios included: using non-adjusted prevalences of obesity in the population, proportionally adjusting the incidence of non-communicable diseases associated with high body mass index considering the differences between Brazil and the United States of America using data from the Global Burden of Disease (GBD) Study [29] and altering the estimated percentage of additional costs of hospitalizations in ±5%.

Further details of the parameters for the estimations and the modeling methodology are described in the Supplementary Materials.

**Table 2. Childhood and adolescent obesity trends in Brazil from 2013 to 2022, by age-group.**

| Age-group | Prevalence 2013 | Prevalence 2022 | Annual % increment rate (95% CI) | p-value | Situation |
|---|---|---|---|---|---|
| **0 to 4 years** | 8.46% | 6.63% | -2.969% (-4.802%-1.136%) | 0.006 | Decreasing |
| **5 to 9 years** | 11.92% | 16.21% | 3.033% (1.087%-4.980%) | 0.007 | Increasing |
| **10 to 19 years** | 5.77% | 12.78% | 6.484% (5.054%-7.914%) | <0.001 | Increasing |

## Results

According to the Brazilian health information systems, obesity has different trends among children and adolescents according to their age-group (Table 2). For children under 5 years of age, the prevalence of obesity showed a small reduction from 2013 to 2022 (2.97% decrease per year), starting at 8.46% in 2013 and reaching 6.63% in 2022. The prevalence of obesity increased from 11.92% to 16.21% among children of 5 to 9 years of age (3.03% increase per year) and from 5.77% to 12.78% among adolescents (6.48% increase per year).

According to information from the National Hospital Information System, show in Table 3, obesity is rarely considered as a primary cause of hospitalization of children of 1 to 9 years. For adolescents, the total number and the costs of hospitalizations having obesity as a primary cause is higher, especially for those aged 15 to 19 years (95% of the hospitalizations, costing Int$2.57 million). For this age group, the number of hospitalizations and their corresponding costs gradually increase and peak in 2018–2019 (reaching close to Int$406 thousand per year), but the number of hospitalizations and their costs decreased significantly in 2020 and 2021 and start to increase again in 2022.

The hospitalization costs with children and adolescents with obesity totaled Int$718.52 million over the last decade (CI 95% Int$629.20–807.84 million) and the additional costs compared to individuals with adequate BMI totaled Int$101.56 million (CI 95% Int$88.93–114.18 million) from 2013 to 2022 (Table 4).

During the last decade, the total non-hospital healthcare costs paid by the families, the outpatient costs and the costs of medications were estimated at a total of Int$43.20 million CI 95% Int$37.83–48.57 million), of which Int$6.02 million (CI 95% Int$5.28–6.77 million) represented additional costs attributable to childhood obesity (Tables 5 and 6). Among these costs,

**Table 3. Total number and costs of hospitalizations with obesity as a primary cause in the Brazilian National Health System (SUS) (Int$), Brazil 2013–2022.**

| Idade | 2013 | 2014 | 2015 | 2016 | 2017 | 2018 | 2019 | 2020 | 2021 | 2022 |
|---|---|---|---|---|---|---|---|---|---|---|
| **1 to 4 years** | | | | | | | | | | |
| **Number** | 0 | 1 | 1 | 0 | 0 | 1 | 0 | 1 | 0 | 1 |
| **Costs** | 0 | 34 | 584 | 0 | 0 | 520 | 0 | 23 | 0 | 192 |
| **5 to 9 years** | | | | | | | | | | |
| **Number** | 0 | 0 | 1 | 2 | 3 | 1 | 0 | 3 | 3 | 1 |
| **Costs** | 0 | 0.00 | 28 | 3,000 | 224 | 419 | 0 | 253 | 4,421 | 155 |
| **10 to 14 years** | | | | | | | | | | |
| **Number** | 4 | 2 | 2 | 3 | 2 | 2 | 4 | 2 | 2 | 4 |
| **Costs** | 3,333 | 690 | 3,241 | 6,709 | 3,214 | 3,184 | 792 | 4,800 | 2,396 | 681 |
| **15 to 19 years** | | | | | | | | | | |
| **Number** | 86 | 72 | 100 | 128 | 143 | 143 | 147 | 38 | 24 | 43 |
| **Costs** | 264,541 | 236,139 | 294,646 | 358,571 | 387,932 | 402,333 | 391,264 | 91,414 | 46,921 | 92,952 |

**Table 4. Total costs and estimated attributable costs to obesity of hospitalizations in the Brazilian National Health System (SUS) among children and adolescents (Int$), Brazil 2013–2020.**

|  | Total hospitalization costs | Additional hospitaliztion costs |
|---|---|---|
| **1 to 4 years** | 197,910,034 | 27,973,681 |
|  | (173,307,797–222,513,083) | (24,496,267–31,451,210) |
| **5 to 9 years** | 182,907,985 | 25,853,210 |
|  | (160,170,656–205,646,065) | (22639392–29067134) |
| **10 to 14 years** | 95,567,543 | 13,508,037 |
|  | (83,687,522–107,447,956) | (11,828,850–15,187,280) |
| **15 to 19 years** | 242,131,636 | 34,224,203 |
|  | (212,032,202–272232063) | (29,969,785–38,478,761) |
| Total | 718,517,199 | 101,559,132 |
|  | (629,198,176–807,839,167) | (88,934,295–114,184,385) |

the healthcare costs paid by the families represent 47,6% of these additional costs (R$2.91 million).

Therefore, considering the sum of total hospital, non-hospital, outpatient, and medication costs with children and adolescents with obesity was estimated at almost Int$761.71 billion (CI 95% Int$667.03–856.41 million) from 2013 to 2022, and the additional costs attributable to childhood obesity totaled approximately Int$107.58 million (CI 95% Int$94.21–120.96 million) during the last decade.

In the sensitivity analysis, the estimated with different parameters generated small differences, between -3% and +3%, when compared to the results of the primary model.

## Discussion

This study firstly investigated the trends in childhood and adolescent obesity and the estimated healthcare costs with children and adolescents with obesity. During this period, the prevalence of obesity has decreased 3.0% per year among children under 5 years of age monitored by primary healthcare services, but it has increased significantly among of those aged 5 to 9 years (3.0% per year) and 10 to 19 years (6.5% per year).

The results for adolescents are consistent with the existing national surveys, which have reported a gradual increase in adolescent obesity over the last decades [30,31], while there is a lack of more frequent recent national surveys for children during this period to compare our time trends. However, prevalence of obesity according to a 2019 survey for children under 5

**Table 5. Estimated non-hospital, outpatient and medication costs of children and adolescents with obesity, Brazil 2013–2020.**

|  | Non-medical costs | Outpatient costs | Medication costs | Total |
|---|---|---|---|---|
| 1 to 4 years | 5,663,562.20 | 1,587,210.30 | 4,647,487.87 | 11,898,260.37 |
|  | (4,959,523.6–6,367,624.01) | (1,389,903.86–1,7845,23.25) | (4,069,757.69–5,225,237.11) | (10,419,185.15–13,377,384.37) |
| 5 to 9 years | 5,234,250.77 | 1,466,896.00 | 4,295,197.29 | 10,996,344.06 |
|  | (4,583,579.97–5,884,943.03) | (1,284,545.85–1,649,252.16) | (3,761,260.42–4,829,151.76) | (9,629,386.24–12,363,346.95) |
| 10 to 14 years | 2,734,842.25 | 766,438.09 | 2,244,196.45 | 5,745,476.79 |
|  | (2,394,873.44–3,074,822.27) | (671,162.01–861,717.31) | (1,965,219.93–2,523,182.18) | (5,031,255.38–6459721.76) |
| 15 to 19 years | 6,929,045.22 | 1,941,861.25 | 5,685,936.26 | 14,556,842.73 |
|  | (6,067,694.7,790,424.68) | (1,700,468.07–2,183,262.38) | (4,979,116.34–6,392,779.49) | (12,747,278.59–16,366,466.55) |
| Total | 20,561,700.44 | 5,762,405.63 | 16,872,817.87 | 43,196,923.95 |
|  | (18,005,671.19–23,117,813.99) | (5,046,079.79–6,478,755.1) | (14,775,354.38–18,970,350.54) | (37,827,105.36–48,566,919.63) |

**Table 6. Estimated additional non-hospital, outpatient and medication costs of children and adolescents with obesity for families and the Brazilian National Health System (SUS), Brazil 2013–2020 (average and 95% Confidence Intervals).**

|  | Non-medical costs | Outpatient costs | Medication costs | Total |
|---|---|---|---|---|
| 1 to 4 years | 800,518.70 | 202,112.56 | 656,901.23 | 1,659,532.49 |
|  | (701,006.05–900,034.63) | (176,987.91–227,238.04) | (575,241.7–738,563.45) | (1,453,235.66–1,865,836.12) |
| 5 to 9 years | 739,837.50 | 186,791.95 | 607,106.57 | 1,533,736.02 |
|  | (647,868.15–831,809.89) | (163,571.81–210,012.86) | (531,637.02–682,578.6) | (1,343,076.98–1,724,401.35) |
| 10 to 14 years | 386,557.50 | 97,596.88 | 317,206.95 | 801,361.34 |
|  | (338,504.46–434,612.13) | (85,464.59–109,729.57) | (277,774.89–356,640.32) | (701,743.94–900,982.02) |
| 15 to 19 years | 979,388.97 | 247,273.19 | 803,681.18 | 2,030,343.33 |
|  | (857,640.92–1,101,141.03) | (216,534.61–278,012.78) | (703,775.4–903,590.25) | (1,777,950.93–2,282,744.06) |
| Total | 2,906,302.67 | 733,774.58 | 2,384,895.93 | 6,024,973.18 |
|  | (2,54,5019.58–3,267,597.67) | (642,558.91–824,993.26) | (2,088,429.02–2,681,372.61) | (5,276,007.51–6,773,963.54) |

years of age [32]was almost half of that found for children accompanied by primary health care services in our study, demonstrating that children that depend on the National Health System are likely to be more vulnerable to the double burden of malnutrition.

Regarding the healthcare cost analysis, the hospitalizations of children and adolescents with obesity as a primary cause totaled Int$2.6 million to the Brazilian National Health System from 2013 to 2022, mostly among adolescents from 15 to 19 years. This demonstrates that obesity is rarely considered as a cause of hospitalization especially among children and adolescents.

Meanwhile, the additional costs of hospitalizations attributable to childhood obesity totaled Int$101.56 million during the same period. Considering the additional non-hospital, outpatient and medication cost attributable to childhood obesity in Brazil, the total additional healthcare costs attributable to obesity were estimated at Int$107.58 million in the last decade. As a comparison, from 2013 to 2022, the total hospitalization costs of children and adolescents from all causes in chapter 4 of the International Code of Diseases (ICD-10), which encompasses endocrine, nutritional, and metabolic diseases, including malnutrition, micronutrient deficiencies, diabetes, and obesity), totaled Int$71.7 million.

These estimates demonstrate that the additional costs attributable to childhood obesity represent a large burden to the National Health System and to the families and, considering the current trends, they tend to increase in the future, together with the added risks of obesity and NCDs in adulthood, which will generate an even larger direct and indirect costs to the country. In 2018, the attributable costs of obesity the National Health System among adults were estimated at Int$637 million [23]and future projections indicate that the direct and indirect costs of obesity may represent up to 4.66% of the Gross Domestic Product in 2060, totaling US $218.2 billion [33].

Healthcare costs vary among the Brazilians, according to factors such as household income, location, race, sex, and age, and approximately 70% of the population depends on the National Health System, with higher coverages among low-income households, rural communities, and among the black population [34]. The out of pocket healthcare expenditures of people with obesity and overweight are higher than those with adequate BMI, because they spend more on medications, medical insurance, exams, and medical consultations, and have been estimated at R$249 (Int$192) per family monthly in 2008–2009 [35].

Additionally, if the current trends of increase in overweight and obesity in Brazil are maintained until 2030, their prevalences among adults may reach 68.1% and 29.6%, respectively [36]. This growth is estimated to result in 5.26 million new cases and 808 thousand deaths

from outcomes such as cardiovascular diseases, diabetes, cancers and chronic kidney disease until the end of the decade [37] and represent almost Int$2 billion in terms of direct healthcare costs and Int$20 billion in losses due to premature deaths in the country[38]. Although the prevalence of non-communicable diseases tends to be higher among black people compared to the white, no studies have assessed these inequities in association with obesity in Brazil [39–41].

These trends are strongly associated with the nutritional transition in Brazil, in which traditional diets, based on foods as rice, beans, fruits and vegetables are gradually being replaced by ultra-processed products in all age groups, including children and adolescents [42]. According to the National Study on Child Food and Nutrition (Enani 2019), some 80% of children under 2 years of age have already consumed ultra-processed foods, which represents a premature exposure to unhealthy foods and does not follow the recommendations of the Brazilian Dietary Guidelines for Children Under 2 Years[43].

Similarly, the national studies on adolescent health, such as the National School Health Survey, which evaluated adolescents from 13 to 17 years old in 2019, observed that although the consumption of sugar sweetened beverages slightly reduced compared to 2015, there is a very high frequency of consumption of ultra-processed foods and low consumption of fruits among the students [44].

Also concerning, the results from the 2017–2018 Household Budget Survey (POF) show that the participation of ultra-processed foods in the total energy of the diets of adolescents is even higher that of the adults. i.e. 26.7% compared to 19.5% of the total calories of the diet [45].

There are other equity concerns in this regard, considering that the prevalences of obesity and the consumption of ultra-processed foods have increased more rapidly among lower income families [46,47]. These socioeconomic concerns also impact of food availability and prices, because, since the Covid-19 pandemic, the food patterns based on ultra-processed foods has become cheaper than the fresh and minimally processed foods, such as fruits, vegetables and whole grains [48]. Additionally, with the loss of income and jobs during the pandemic, many people abandoned supplementary health coverage to use the public health system.

Notably, the consumption of ultra-processed foods has been associated with higher risk of overweight and obesity [49]and of other NCDs, including cardiovascular diseases, diabetes and some types of cancers [50–52]. Also, the consumption of added sugar by Brazilians is high in all age-groups and requires effective public policy responses [45]. Therefore, multicomponent and intersectoral strategies are required to reduce and control obesity in all age groups, especially among children and adolescents, considering regulatory and fiscal policies, nutritional education, healthy food procurement and other policies.

The results of this study demonstrate the need for strengthening and expanding the control and prevention of childhood obesity in Brazil, considering more cost-effective interventions for treating overweight and obesity and promoting regulatory and fiscal policies that promote heathy dietary environments during all phases of childhood and adolescence, from promoting breastfeeding and complementary feeding to regulating school canteens, regulating food marketing, improving nutritional labeling, taxing ultra-processed foods and beverages and creating subsidies for fresh and minimally processed foods [53–55]. In parallel, education on healthy diets should be provided for all age groups based on the recommendations of the existing national dietary guidelines [56,57].

The strengths of this study include using a modeling method that was based on primary studies from a recent meta-analysis on the costs of childhood obesity and overweight in different countries and the use of updated data on nutritional status and costs of hospitalizations from national health information systems. The results in this study are proportionally lower than estimated additional costs of hospitalizations attributable to childhood obesity [21],

because we assumed conservative parameters in the modeling and costs were mostly limited to the National Health System and does not include the costs of primary healthcare services. Also, if the costs of overweight were included in the estimations, the additional costs would be even higher.

The limitations of the study include the use of estimates of hospitalization costs from high-income countries for the modeling, the possible coverage and quality limitations of the national data sources, and not including primary healthcare costs and costs from the supplementary (private) health services in Brazil.

However, the assumptions that were adopted in this study allowed the first conservative estimation of direct costs of hospitalizations attributable to childhood obesity in Brazil. In the future, new studies will be important for direct and more precise assessment of these costs using data linkage of information from the national health information systems and possibly information from national childhood cohorts.

In conclusion, the results of this study highlight that the prevalence of childhood and adolescent obesity is increasing in most age-groups and that their costs are not limited to the impacts on adult health and represent a relevant economic burden to the Brazilian National Health System and to families because of additional healthcare costs during childhood. Therefore, the prevention and control of childhood and adolescent obesity are public health priorities that demand effective policies.

## Supporting information

**S1 Table. Data from the primary studies selected for the estimation of the percentage of additional hospitalization costs attributable to childhood obesity (sample size, percentage of additional costs per study, author and year of publication, country of the study, and age-group studied) and the final estimates.**
(DOCX)

**S2 Table. Total costs of hospitalizations (Int$) from all causes of children and adolescents by age-group from 2013 to 2022 (National Hospital Information System—SIH/SUS).**
(DOCX)

**S3 Table. Average per capita costs of hospitalizations (Int$) from all causes of children and adolescents by age-group from 2013 to 2022 (National Hospital Information System—SIH/SUS).**
(DOCX)

**S4 Table. Total number of hospitalizations from all causes of children and adolescents by age-group from 2013 to 2022 (National Hospital Information System—SIH/SUS).**
(DOCX)

**S5 Table. Original and adjusted prevalence of obesity by age-group from 2013 to 2022 (Sisvan).**
(DOCX)

**S6 Table. Total number of individuals monitored by Sisvan from 2013 to 2022, by age-group.**
(DOCX)

**S7 Table. Proportion of non-hospital, outpatient and medication costs in relation to the hospitalization costs from Ling et al, 2023 [21].**
(DOCX)

**S1 Fig. Sensitivity analysis comparing the primary model to different data input scenarios.** (DOCX)

## Author Contributions

**Conceptualization:** Eduardo Augusto Fernandes Nilson.

**Data curation:** Eduardo Augusto Fernandes Nilson.

**Formal analysis:** Eduardo Augusto Fernandes Nilson.

**Investigation:** Eduardo Augusto Fernandes Nilson.

**Methodology:** Eduardo Augusto Fernandes Nilson.

**Supervision:** Eduardo Augusto Fernandes Nilson, Ana Carolina Rocha de Oliveira, Raphael Barreto da Conceição Barbosa.

**Writing – original draft:** Eduardo Augusto Fernandes Nilson.

**Writing – review & editing:** Eduardo Augusto Fernandes Nilson, Michele Gonçalves da Costa, Ana Carolina Rocha de Oliveira, Olivia Souza Honorio, Raphael Barreto da Conceição Barbosa.

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
