## [Editor Report · Decision Letter 0]

23 Nov 2023

PONE-D-23-37832Estimation of the direct health costs attributable to child obesity in Brazil from 2013 to 2022PLOS ONE

Dear Dr. de Oliveira ,

Thank you for submitting your manuscript to PLOS ONE. After careful consideration, we feel that it has merit but does not fully meet PLOS ONE’s publication criteria as it currently stands. Therefore, we invite you to submit a revised version of the manuscript that addresses the points raised during the review process.

We look forward to receiving your revised manuscript.

Kind regards,

Alanna Gomes da Silva, PhD

Academic Editor

PLOS ONE

Journal Requirements:

   "I have read the journal's policy and the authors of this manuscript have the following competing interests: EAFN is currently an Academic Editor for PLOS One."

Additional Editor comments: 

The work entitled "Estimation of the direct health costs attributable to child obesity in Brazil from 2013 to 2022" is of great relevance to public health, especially considering the nutritional transition and the obesity epidemic affecting almost everyone, with an alarming impact on children and adolescents. The article is written with clarity and scientific rigor, but after careful reading, I made some suggestions on specific points in the text that require corrections, especially in the methods. After review, the manuscript may be accepted.

INTRODUCTION

Update references 1, 2, and 3 as the data is quite old (2009 for Brazil and 2016 worldwide) and may not represent the current scenario.

The authors state that there are no studies on the cost of childhood obesity to the SUS (Brazil's Unified Health System). Did the authors conduct a literature review to obtain this information?

The study's objective is "...to estimate the direct healthcare costs...," but the introduction does not mention the meaning of "direct healthcare costs." It is important to define this term for a better understanding of the topic.

Another issue regarding the objectives: the methods and results present information beyond costs, as the trend of obesity is analyzed. Therefore, the study will have two objectives: the first is to analyze the trend and the second is to estimate direct healthcare costs. I suggest adding this objective to make it consistent with the results. Also, objectives should be included in the abstract.

METHODS

The methods need to be clearer and more complete, following the checklist for observational studies (STROBE), which should include the following information: Study design; Setting; Participants; Variables; Data sources/measurement; Bias; Study size; Quantitative variables; Statistical methods. Available at: https://www.strobe-statement.org/checklists/

I also suggest that the part about trend analysis be more detailed, explaining how this variable is presented by Sisvan, if there is the number of children and adolescents in the system each year. Did the authors consider using the Prais-Winsten regression?

The authors work with Average Annual Percent Change but did not consider the regression p-value to determine if the change was significant. I suggest adding this and including it in the results.

Another issue that needs justification is the age cutoff of up to 19 years. According to the Brazilian Statute of the Child and Adolescent, a person is considered a child up to twelve incomplete years of age, and an adolescent is between twelve and eighteen years of age.

DISCUSSION

The authors correctly start the discussion by summarizing the results. However, the trend is not mentioned. If it is in the results, it cannot be ignored. The authors discussed the increasing trend of obesity due to nutritional transition, so it is important to mention this result in the first paragraph and follow the sequence presented in the results: first, the trend, and then the costs.

The authors present data from studies showing adolescents' dietary issues but do not mention the National School Health Survey, which provides this information and is representative of the Brazilian population.

CONCLUSION

Review the conclusion, as it should strictly address the study's objectives. Therefore, the main findings of the study should be added initially, also altering the abstract.

We look forward to receiving your revised manuscript.

---

## [Author Response · Author response to Decision Letter 0]

15 May 2024

Thank you for the comments and suggestions to the original manuscript, especially regarding the regression methodology that would be preferable for the analysis. We have revised the manuscript accordingly.

---

## [Decision Letter · Decision Letter 1]

16 Jun 2024

PONE-D-23-37832R1Estimation of the direct health costs attributable to child obesity in Brazil from 2013 to 2022PLOS ONE

Dear Dra. de Oliveira,

Thank you for submitting your manuscript to PLOS ONE. After careful consideration, we feel that it has merit but does not fully meet PLOS ONE’s publication criteria as it currently stands. Therefore, we invite you to submit a revised version of the manuscript that addresses the points raised during the review process.

We look forward to receiving your revised manuscript.

Kind regards,

Alanna Gomes da Silva, PhD

Academic Editor

PLOS ONE

Additional Editor Comments:

Dear Oliveira,

After the initial review, other reviewers were invited to review the manuscript and we sent it for corrections.

After careful review, we believe that the manuscript has the potential for publication, however, it will require " Major Revision" throughout the text. I request that you respond to all inquiries in a separate document and that all changes in the text be highlighted in a different color. Consider the latest corrected version submitted by the authors in May 2024.

Several issues need to be addressed: Additionally, the manuscript is not formatted according to the citation and reference style required by the journal, which adopts the "Vancouver Style." Therefore, citations should be numerical and superscripted after the period at the end of each paragraph. Please carefully review the "Submission Guidelines" criteria, available at: https://journals.plos.org/plosone/s/submission-guidelines#loc-manuscript-organization, and make all necessary corrections.

Reviewers' comments:

Reviewer's Responses to Questions

**Comments to the Author**

1. If the authors have adequately addressed your comments raised in a previous round of review and you feel that this manuscript is now acceptable for publication, you may indicate that here to bypass the “Comments to the Author” section, enter your conflict of interest statement in the “Confidential to Editor” section, and submit your "Accept" recommendation.

Reviewer #1: All comments have been addressed

Reviewer #2: All comments have been addressed

2. Is the manuscript technically sound, and do the data support the conclusions?

Reviewer #1: Yes

Reviewer #2: Yes

3. Has the statistical analysis been performed appropriately and rigorously? 

Reviewer #1: Yes

Reviewer #2: Yes

4. Have the authors made all data underlying the findings in their manuscript fully available?

Reviewer #1: Yes

Reviewer #2: Yes

5. Is the manuscript presented in an intelligible fashion and written in standard English?

Reviewer #1: Yes

Reviewer #2: Yes

6. Review Comments to the Author

Reviewer #1: Exclude the excerpt below from the results and include in the methodology:

For the estimation of the additional percentage of per capita hospitalization costs, we

selected 24 of the primary studies comparing children and adolescents with obesity and

Those with adequate BMI primary from the meta-analysis by Ling et al, 2023. The

studies had variable sample sizes and included children and adolescents from 0 to 19

years of age, and the estimated percentage of additional hospitalization costs of children and adolescents with obesity was of 16.46% (CI 95%: 1.98%-30.94%).

Include in the results: How many hospitalizations were retrieved from the National Hospital Information System SIH/SUS, in the period from 2013 to 2022, for each age group (1 to 4 years, 5 to 9 years, 10 to 14 years and 15 to 19 years) ?

Reviewer #2: Thank you for the opportunity to review the article and learn from the authors. Below I consider some points, which I consider key to aligning the article and bringing greater clarity. Some questions still need improvement for greater reader understanding.

I suggest following the strobe.

The title of the study must be connected to the objective. It is not mentioned in the title that the study covers adolescents.

Summary

I suggest including the objective of the study in the introduction of the abstract. It is important that the objective is mentioned so that the reader understands whether the methods adopted achieve these objectives.

Insert the metrics used in the methods.

Introduction

The authors mention gaps related to child surveys, which actually happens. However, for adolescents we have an important survey, the National School Health Survey, which also measures risk and protective factors for chronic non-communicable diseases, including obesity. I think it is essential to bring this information.

Another issue that I suggest should be better addressed is the problem of the study. Furthermore, how does the study advance and contribute in relation to the others?

Methods

On page 10, line 85, and in places where the authors mention the Brazilian National Health System, I suggest that it be included in parentheses with the nomenclature of the SUS (Sistema Único de Saúde – SUS in Portuguese), since it is the form used in the country.

What information and reporting systems were used? It is important to mention them from the beginning so that the reader understands the data sources used. These sources are mentioned below, I suggest mentioning them all from the beginning and then bringing the source used in detail in each subtopic. It becomes clearer for the reader.

What references did the authors use to define children and adolescents?

On page 11, line 107, I suggest including the trend of obesity in adolescents in the subtopic, as it covers the age range from 10 to 19 years old.

What was the sample studied?

What were the inclusion and exclusion criteria for the study.

On page 40, line 224, the authors mention that adjustments were made for the incidence of chronic non-communicable diseases, was it possible to obtain this information?

Results

Which final sample made up the analyzed strata.

The results in the first paragraph present obesity trends in children and adolescents, but this does not appear in the objective. This way it becomes disconnected. I suggest that it be included in the title, objective and aligned throughout the text.

Prevalence confidence intervals are not presented. Were they calculated?

Align the title of table 2 with the inclusion of the adolescent population and the sources of information used.

Tables 3, 4, 5 and 6 include the population studied and the sources of information used.

In the results, it is not possible to discuss the findings with the COVID-19 pandemic, I suggest that this appears in the discussion, as the pandemic was not even mentioned in the introduction. The results must follow the objective and methods. This can be found in the presentation of results in table 3.

Discussion

In the first paragraph it is mentioned that obesity trends in the target population were studied, but this does not appear in the objective as mentioned previously.

The conclusion needs to respond to the objective.

I suggest that the limitations and then the strengths of the study are presented in a sequential manner. On page 21, on line 430, start with the limitations, then talk about the strengths and continue with the paragraph on the conservative approach to calculating costs.

In the discussion it is important that the authors bring information from the National School Health Survey, which has already had 4 editions and the National Health and Demographic Survey, even with the last edition in 2006, it brought information about children and this research is in the field. These are important surveys with children and adolescents that also address topics related to the content of the article, they are the broadest surveys with this population that we have at a national level.

The discussion is fragile, I suggest that the results of this study be discussed in more detail. What explain the significant results in the tables? How, for example, does obesity decrease between 0 and 4 years old and increase in other age groups? In the discussion nothing was mentioned about the COVID-19 pandemic, I think it would be interesting to bring up what they mentioned about it in the results and discuss it.

Why do they assume that hospital costs were lower than literature data that did not include the supplementary health system? This needs to be better explained. What is the reason?

Furthermore, obesity is also related to availability, as well as food choice, ultra-processed foods tend to be more affordable and can lead to obesity, on the other hand, vegetables and fruits are more expensive. This needs to be discussed in the various socioeconomic strata because it will lead to obesity, whether in those who use the SUS or those who use supplementary healthcare. Furthermore, after the pandemic, many people lost their health insurance, and there is the duality of obesity and access to food and malnutrition and hunger. I suggest discussing this.

7. PLOS authors have the option to publish the peer review history of their article (what does this mean?). If published, this will include your full peer review and any attached files.

Reviewer #1: **Yes: **TÉRCIA MOREIRA RIBEIRO DA SILVA

Reviewer #2: **Yes: **Ana Carolina Micheletti Gomide Nogueira de Sá

---

## [Author Response · Author response to Decision Letter 1]

22 Jul 2024

As requested by the review letter, we are sending you the files “Response to Reviewers”, “Manuscript” and “Revised Manuscript with Track Changes”.

---

## [Editor Report · Decision Letter 2]

25 Jul 2024

Estimation of the direct health costs attributable to child obesity in Brazil from 2013 to 2022

PONE-D-23-37832R2

Dear Dr. Oliveira,

We’re pleased to inform you that your manuscript has been judged scientifically suitable for publication and will be formally accepted for publication once it meets all outstanding technical requirements.

Kind regards,

Alanna Gomes da Silva, PhD

Academic Editor

PLOS ONE
---

## [Editor Report · Acceptance letter]

11 Sep 2024

PONE-D-23-37832R2 

PLOS ONE

Dear Dr. de Oliveira, 

I'm pleased to inform you that your manuscript has been deemed suitable for publication in PLOS ONE. Congratulations! Your manuscript is now being handed over to our production team.

Kind regards, 

on behalf of

Dr. Alanna Gomes da Silva 

Academic Editor

PLOS ONE